# SKGQA, a Peptide Derived from the ANA/BTG3 Protein, Cleaves Amyloid-β with Proteolytic Activity

**DOI:** 10.3390/biom14050586

**Published:** 2024-05-15

**Authors:** Yusuke Hatakawa, Rina Nakamura, Toshifumi Akizawa, Motomi Konishi, Akira Matsuda, Tomoyuki Oe, Motoaki Saito, Fumiaki Ito

**Affiliations:** 1Department of Bio-Analytical Chemistry, Graduate School of Pharmaceutical Sciences, Tohoku University, 6-3 Aramaki-Aoba, Aoba-ku, Sendai 980-8578, Miyagi, Japan; yusuke.hatakawa.e6@tohoku.ac.jp (Y.H.); t-oe@mail.pharm.tohoku.ac.jp (T.O.); 2O-Force Co., Ltd., 3454 Irino Kuroshio-cho, Hata-gun 789-1931, Kochi, Japan or r.nakamura@kochi-u.ac.jp (R.N.); or jm-momizit@kochi-u.ac.jp (T.A.); 3Department of Pharmacology, Kochi Medical School, Kochi University, Kohasu, Oko-cho, Nankoku 783-8505, Kochi, Japan; saitomo@kochi-u.ac.jp; 4Department of Integrative Pharmacy, Faculty of Pharmaceutical Sciences, Setsunan University, 45-1 Nagaotoge-cho, Hirakata 573-0101, Osaka, Japan; motomi@pharm.setsunan.ac.jp; 5Laboratory of Medicinal and Biochemical Analysis, Faculty of Pharmaceutical Sciences, Hiroshima International University, 5-1-1 Hirokoshinkai, Kure 737-0112, Hiroshima, Japan; matsuda@hirokoku-u.ac.jp; 6The Institute of Prophylactic Pharmacology, 1-58, Rinku-oraikita, Izumisano 598-8531, Osaka, Japan

**Keywords:** Catalytide, proteolytic peptide, ANA-SA5, ANA-TA9, JAL-TA9, Alzheimer’s disease, amyloid-β, neurodegenerative disease, peptide drug, serine proteases

## Abstract

Despite the extensive research conducted on Alzheimer’s disease (AD) over the years, no effective drug for AD treatment has been found. Therefore, the development of new drugs for the treatment of AD is of the utmost importance. We recently reported the proteolytic activities of JAL-TA9 (YKGSGFRMI) and ANA-TA9 (SKGQAYRMA), synthetic peptides of nine amino acids each, derived from the Box A region of Tob1 and ANA/BTG3 proteins, respectively. Furthermore, two components of ANA-TA9, ANA-YA4 (YRMI) at the C-terminus end and ANA-SA5 (SKGQA) at the N-terminus end of ANA-TA9, exhibited proteolytic activity against amyloid-β (Aβ) fragment peptides. In this study, we identified the active center of ANA-SA5 using AEBSF, a serine protease inhibitor, and a peptide in which the Ser residue of ANA-SA5 was replaced with Leu. In addition, we demonstrate the proteolytic activity of ANA-SA5 against the soluble form Aβ42 (*a*-Aβ42) and solid insoluble form *s*-Aβ42. Furthermore, ANA-SA5 was not cytotoxic to A549 cells. These results indicate that ANA-SA5 is a promising Catalytide and a potential candidate for the development of new peptide drugs targeting Aβ42 for AD treatment.

## 1. Introduction

Our previous studies on a short hydrolytic peptide named Catalytide present an attractive possibility for the development of new strategic drugs for the treatment of Alzheimer’s disease (AD) [1,2,3,4,5,6,7].

AD is the most prevalent age-related neurodegenerative disorder globally, and the use of an effective drug over a long period of time is required. Despite many studies targeting amyloid-β (Aβ) 42 degradation, clearance, inhibition of aggregation, or oligomerization, all strategies have failed in the clinical setting [8,9,10,11]. Thus, it is not certain whether the Aβ cascade is the main cause of AD. Recently, the antibody drugs lecanemab and aducanumab have been recognized as effective AD drugs, giving great hope to patients with AD [12]. The main disadvantages of antibody drugs are the side effects, which are thought to be caused by the deposition of Aβ removed by anti-amyloid antibodies in the blood and lymphatic vessels [13,14,15]. Thus, new strategies are required.

In our previous work, we reported the discovery of a novel hydrolase peptide, JAL-TA9, consisting of nine amino acids derived from the Box A region of Tob1. Its proteolytic activity was inhibited by 4-(2-aminoethyl) benzenesulfonyl fluoride hydrochloride (AEBSF), a serine protease inhibitor [5]. We named this proteolytic peptide a Catalytide (catalytic peptide), and, to our knowledge, this is the first report of a hydrolase peptide. Surprisingly, JAL-TA9 showed proteolytic activity and cleaved Aβ42 [3]. The Tob/BTG family of proteins is involved in the cell cycle and the regulation of a variety of cells, such as T lymphocytes, fibroblasts, epithelial cells, and germ cells [16,17,18,19,20], and contains three highly conserved regions, Box A, Box B, and Box C, at the N-terminus. Despite numerous reports on the Tob/BTG family proteins, the functions of these three regions are not well-understood. Therefore, we investigated the proteolytic activity of ANA-TA9 (SKGQAYRMI) derived from the ANA/BTG3 protein [6]. This peptide cleaved three types of Aβ fragment peptides, Aβ1—20, Aβ11—29, and Aβ28—42. Aβ11—29 is the fragment peptide derived from the central region, thought to be the core region for the aggregation and oligomerization of Aβ42. ANA-TA9 also cleaved not only the authentic soluble form Aβ42 (*a*-Aβ42) but also the solid insoluble form Aβ42 (*s*-Aβ42) at many cleavage sites, especially in their central regions in a similar way to JAL-TA9 [3,21]. Therefore, we concluded that ANA-TA9 is a Catalytide similar to JAL-TA9.

An analysis of the autoproteolytic activity of ANA-TA9 using high-performance liquid chromatography (HPLC) every hour for eight hours of incubation showed that the chromatogram pattern changed continuously, even after ANA-TA9 disappeared due to autolysis (Figure 1a). These results suggest that the C-terminal peptide fragment of ANA-TA9 (ANA-YA4; YRMI), which appeared because of the autolysis of ANA-TA9 and was subsequently reduced, may have proteolytic activity (Figure 1a). On the other hand, it has been found that the active center of ANA-TA9 is the N-terminal Ser. These results suggested that not only ANA-YA4 but also ANA-SA5 containing N-terminal Ser may have proteolytic activity. Therefore, we synthesized ANA-YA4 and its N-terminus, SKGQA (ANA-SA5), and evaluated their proteolytic activity (Figure 1b). Both peptide fragments YRMI and SKGQA cleaved Aβ1—20, Aβ11—29, and Aβ28—42 fragment peptides of Aβ42, similar to ANA-TA9. In addition, kinetic parameters showed that, among ANA-TA9, ANA-SA5, and ANA-YA4, ANA-SA5 has the highest affinity for Aβ11—29, which is known to form β-sheets and contains the regions essential for the oligomerization and aggregation of Aβ42 [6]. These data suggest that ANA-SA5 is a better seed peptide for AD treatment.

The absorption, disposition, and brain delivery of Catalytides should be determined for their clinical application in the treatment of AD. Previously, we revealed that ANA-TA9 is directly delivered to the brain via the olfactory system by nasal administration [7]. It has been reported that the lower the membrane permeability, the higher the amount transferred to the olfactory bulb-trigeminal nerve, and the higher the efficiency of delivery to the brain during nasal administration [22]. Generally, membrane permeability is largely influenced by the degree of hydrophobicity of a substance, and highly hydrophilic substances have a low membrane permeability. Therefore, the hydrophobicity of ANA-TA9, ANA-SA5, and ANA-YA4 was calculated using the grand averages of the hydropathicity parameters. As a result, ANA-TA9, ANA-SA5, and ANA-YA4 were −0.689, 0.15, and −1.360, respectively. Therefore, we predicted that ANA-SA5 has the highest transferability to the brain. Based on these findings, ANA-SA5 is considered an appropriate candidate for clinical use in AD therapy.

Here, we investigated the potential of ANA-SA5 as a therapeutic agent for AD. In this study, we identified the active center of ANA-SA5 using structural analysis, in addition to studies using inhibitors and mutants with amino acid substitutions. Next, we identified the proteolytic activity of ANA-SA5 against both *a*-Aβ42 and *s*-Aβ42.

## 2. Materials and Methods

### 2.1. Preparation of Peptides

Peptides were synthesized based on our prior work, utilizing Fmoc-protected l-amino acid derivatives and an automated peptide synthesizer (model 433A, Applied Biosystems, Waltham, MA, USA., 0.1 mmol scale with preloaded resin) [5]. Following deprotection according to the manufacturer’s instructions, purification was carried out using reversed-phase HPLC (Capcell Pak C18 column, SG, 10 or 15 mm i.d. × 250 mm; Shiseido Co., Ltd., Tokyo, Japan) with a linear elution gradient from 0.1% trifluoro acetic acid (TFA) to 50% or 70% CH_3_CN containing 0.1% TFA over 30 min or an isocratic mode by using 0.1% TFA (ANA-SA5). The flow rate was set at 3 or 6 mL/min. Principal peak fractions were collected and subjected to lyophilization. The purity of the synthetic peptides and the progress of the enzymatic reaction were confirmed by analytical reversed-phase HPLC (Capcell Pak C18 column, MGII, 4.6 mm i.d. × 150 mm; Shiseido Co., Ltd., Tokyo, Japan) at a flow rate of 1.0 mL/min with a linear elution gradient from 0.1% TFA to 70% CH_3_CN containing 0.1% TFA. The column eluate was monitored using a photodiode array detector (SPD-M20A; Shimadzu, Kyoto, Japan). ANA-SA5 and Aβ11-29 exhibited retention times of 3.5 min and 10.6 min, respectively, on the analytical HPLC. Conversely, the solid type of Aβ42 (*s*-Aβ42) was obtained as a slightly brownish solid material post-lyophilization and remained insoluble in CH_3_CN, CH_3_OH, CH_3_COOH, and DMSO. Hence, this solid material was denoted as solid Aβ42 (*s*-Aβ42) after repetitive washing with CH_3_CN and CH_3_OH [3]. After ANA-SA5 and Aβ11-29 were purified, they were characterized via ESI-Mass (MS) utilizing a Qstar Elite Hybrid LC-MS/MS system (Applied Biosystems Inc., Waltham, MA, USA) (Appendix A). Authentic Aβ42 (*a*-Aβ42) was purchased from the peptide institute (Osaka, Japan).

### 2.2. Analysis of Proteolytic Activity and Determination of Cleavage Sites

We conducted a detailed analysis of the proteolytic activity of ANA-SA5 and the determination of its cleavage sites as described previously [6]. In brief, ANA-SA5 was incubated individually with or without *a*-Aβ42 (final concentration 0.05 mM), *s*-Aβ42 (final concentration 0.05 mM), or Aβ11—29 (final concentration 0.05 mM) in the presence of human serum albumin (HSA) at a final concentration of 0.025% *w*/*v* in PBS (pH 7.4) at 37 °C. Immediately before use, Aβ11—29 and *a*-Aβ42 were dissolved in MilliQ water to 1 mM and in DMSO to 5 mM, respectively. In the case of *s*-Aβ42, since it did not dissolve in the solvent, the reaction was carried out in the buffer solution in solid form. A 10 µL aliquot of the reaction mixture was subjected to time-dependent analysis using the analytical HPLC system described above. A portion of the reaction mixture (20 μL for *a*-Aβ42 and Aβ11—29; 40 µL for *s*-Aβ42) was injected into the analytical HPLC system to determine the cleavage site. The peak fractions were monitored at 220 nm and collected into microtubes. Following lyophilization, an appropriate volume of 36% CH_3_CN containing 0.1% HCOOH was added based on the chromatographic peak height, and the mixture was stirred using an automatic mixer. The cleavage sites were determined by ESI-MS using a flow injection method with 70% CH_3_CN and 0.1% HCOOH on a QStar Hybrid LC-MS/MS system. The flow rate was set at 0.1 mL/min.

The effects of AEBSF, a serine protease inhibitor, on the proteolytic activity of ANA-SA5 were analyzed in the same manner using 60 mM of AEBSF.

### 2.3. Cell Experiments

Cisplatin (CDDP), WST-8 (Cell Counting Kit-8), and the FLAG peptide (DYKDDDDK) were sourced from FUJIFILM Wako Pure Chemical Corporation, Dojindo Laboratories, and Sigma-Aldrich, respectively. The human lung cancer cell line A549 was obtained from Riken Cell Bank (Ibaraki, Japan). Briefly, A549 cells were exposed to 0.2 mM ANA-SA5, 0.2 mM FLAG peptide, and 4 μM CDDP under the same conditions described previously [21]. After a 72 h incubation, the medium was replaced with 110 μL containing WST-8 reagent (10 μL WST-8 reagent and 100 μL DMEM). The cells were further incubated for 1 h, and absorbance was determined at 450 nm with a reference wavelength of 620 nm using a Spectra Max Plus 384 microplate reader (Molecular Devices, Sunnyvale, CA, USA).

### 2.4. Stereo-Structure Analysis

The stereo-structure of ANA-SA5 was examined in a manner consistent with the methodologies outlined in our prior publications [1,5,6]. We utilized the CSC Chem3D UltraTM software (version 17.0., PerkinElmer, Waltham, MA, USA) for computer modeling of ANA-SA5. The solvent radius was set to 1.4 Å, corresponding to the value for water. The initial conditions for the structure of ANA-SA5 involved setting all peptide bonds and dihedral angles to 180°, after which the six atoms constituting the peptide bond were aligned on a single plane, and the bond lengths were established. Subsequent calculations were performed using structural optimization and energy minimization by MM2 and MMFF94 parameters, which include bond length, bond angles, dihedral angles, dipole moments, and van der Waals values [1,5].

## 3. Results

### 3.1. Identification of the Active Center of ANA-SA5

We previously reported that ANA-SA5 derived from ANA-TA9 cleaved Aβ11—29, which contains the core sequence for Aβ42 aggregation. In ANA-TA9, Ser, Arg, and carboxyl groups have been identified as being involved in its activity. However, the amino acids involved in the activity of ANA-SA5 have not yet been identified. First, we investigated whether ANA-SA5 is inhibited by AEBSF, a serine protease inhibitor that inhibits ANA-TA9 activity. A reaction mixture of ANA-SA5 (final concentration 0.2 mM) and Aβ11—29 (final concentration 0.05 mM) was incubated with HSA in PBS (pH 7.4) at 37 °C in the presence or absence of AEBSF. The reaction mixture was loaded onto an analytical HPLC system. In the absence of AEBSF, several peaks were observed on the chromatogram after 1 day of incubation. Furthermore, an analysis of the peaks that appeared on the chromatogram using mass spectrometry (MS) revealed that ANA-SA5 cleaved Aβ11—29 with 10 cleavage sites (Figure 2a,d). In the presence of AEBSF, both ANA-SA5 and Aβ11—29 were initially identified as a single peak for each (Figure 2b). After one day of incubation, all peaks observed in the chromatogram were collected for MS analysis (Figure 2c). Five peaks (β1–β5) were identified as fragment peptides derived from Aβ11—29, and the cleavage sites were summarized (Table 1 and Figure 2d). It was revealed that the number of cleavage sites on Aβ11—29 by ANA-SA5 decreased to 5. These results revealed that AEBSF inhibited ANA-SA5 activity, but not completely. The specific inhibition mechanism of AEBSF is the sulfonylation of the serine residue at the active center of the serine protease. Based on MS analysis, the molecular weights of the peak marked by C were identified as 637.28 Da (Figure 2b; lower). This molecular weight was the same as the predicted molecular weight when AEBSF was combined with ANA-SA5.

Because MS analysis showed that the molecular weights of the peaks marked with an asterisk (*) did not correspond to amino acids or peptides, it was thought that they might be a byproduct of AEBSF. To confirm the binding site of AEBSF to ANA-SA5, peak C in the chromatogram shown in Figure 2 was used for MS/MS analysis (Figure 3). Using the reading method from the C-terminus (indicated by b), a molecular weight of 398.1536 Da was observed for b2. In addition, in the N-terminus reading method (indicated by y), a precursor ion of 673.2831 Da was observed at y5 and 275.1360 Da, the molecular weight of GQA, was observed at y3, with a difference of 398.15. The molecular weight of the AEBSF bound to SK (398.15) was determined using both methods. Furthermore, the molecular weight of AEBSF bound to S (288.06) was also identified. These results showed that the conjugated position of AEBSF was determined on the OH group of Ser residue at the N-terminus end of ANA-SA5 (Figure 3).

This suggested that Ser plays an important role in the proteolytic activity of ANA-SA5. Therefore, we investigated whether the activity of ANA-SA5 could be abolished by replacing Ser (S) with Leu (L), an amino acid without a hydroxyl group. A reaction mixture of ANA-SA5 S/L (final concentration 0.2 mM) in which Ser was replaced with Leu and Aβ11—29 (final concentration 0.05 mM) was incubated with HSA in PBS (pH 7.4) at 37 °C and analyzed using HPLC on days 0 and 3. The chromatograms had no significant differences between days 0 and 3 (Figure 4). These results revealed that the proteolytic activity of ANA-SA5 ceased with substituting the Ser residue with Leu.

Generally, a catalytic triad is necessary for an enzyme to function as a serine protease, with the active center and an oxyanion hole to stabilize binding to the substrate [23]. According to our previous reports, the stereo structure of ANA-SA5 was estimated by molecular dynamics simulations using the MM2 and MMFF94 parameters [1]. As a result, the distances between Ser (-OH) and the N-terminus (-NH_2_) and between Ala (-COOH) and the N-terminus (-NH_2_) were 2.26 Å and 2.03 Å, respectively (Table 2). This indicated that the catalytic triad of ANA-SA5 was formed by the carbonyl oxygen at the C-terminus (-COOH), base at the N-terminus (-NH_2_), and hydroxyl group of serine (Ser) (Figure 5a). As the main NH chains of Lys and side chains (-NH_2_) of Lys are on the same plane, it is thought that they form an oxyanion hole, as shown in Figure 5b.

These data indicate that ANA-SA5 possesses serine-protease-like activity, like ANA-TA9 [6,21].

### 3.2. Proteolysis of Authentic Soluble Form Aβ42 (a-Aβ42) by ANA-SA5

Next, we examined the proteolytic activity of ANA-SA5 against *a*-Aβ42 as previously reported [3,21]. The reaction mixture of ANA-SA5 and *a*-Aβ42 was analyzed for up to 7 days (Appendix A). Initially, ANA-SA5, *a*-Aβ42, and HSA were eluted at 3.5, 12.8, and 13.5 min, respectively (Figure 6a). On day 1, small peaks appeared at approximately 8 min. Interestingly, the chromatogram patterns, especially the comparative heights of the individual peaks, changed up to day 7. Thus, all peaks (β1–15) on day 7 were collected and analyzed by MS (Figure 6b). As a result, 24 types of peptides, including *a*-Aβ42 and one amino acid, were identified as fragment peptides derived from Aβ42 (Table 3). No peptide fragments derived from HSA were identified by HPLC or MS analyses. With *a*-Aβ42 alone, no fragment peptide was identified. These results indicate that ANA-SA5 cleaves *a*-Aβ42, but not HSA like ANA-TA9.

### 3.3. Proteolysis of Solid Form Aβ42 (s-Aβ42) by ANA-SA5

Before analyzing the proteolysis of *s*-Aβ42, we confirmed the purity and stability of a slightly brownish solid material, referred to as *s*-Aβ42, obtained after lyophilization (Figure 7a), which was insoluble in CH_3_CN, CH_3_OH, CH_3_COOH, and DMSO. After continuously washing with CH_3_CN and CH_3_OH to remove the amino-acid-protecting groups and other by-products, *s*-Aβ42 was incubated alone in PBS (pH 7.4) in the presence of HSA at 37 °C. The reaction mixtures were analyzed in the same manner as *a*-Aβ42 described above. *s*-Aβ42 was not initially identified, indicating that *s*-Aβ42 was not soluble in the reaction buffer (Figure 7a). However, the chromatograms changed continuously for up to 7 days (Appendix A). Thus, all the new peaks (b1–b3) that appeared on day 7 were collected and subjected to MS analysis (Figure 7b). Four peptides, GGVVIA, VVIA, GVVIA, and VGGVVIA, were identified from three peak fractions as analogous peptides derived from the C-terminus of Aβ42 (Table 4). These peptides are considered side products of Aβ42 synthesis because they contain Ala residues at their C-termini [3,21]. In addition, the appearance of *s*-Aβ42 did not change in the absence of ANA-SA5 during seven days of incubation, indicating that *s*-Aβ42 was not cleaved in the reaction buffer.

Next, we examined the proteolytic activity of ANA-SA5 on *s*-Aβ42. Initially, ANA-SA5 and HSA were identified; however, several peaks appeared in the chromatogram on day 1. The heights of the new peaks increased in a time-dependent manner (Appendix A). Thus, we collected all appearing peaks (S1 and β1–17) on day 7 (Figure 8), and 19 peptides, including GGVVIA, identified in the reaction mixture of *s*-Aβ42 alone, were identified as fragments derived from *s*-Aβ42 (Table 5). Based on the MS analysis, we determined the cleavage sites on *a*-Aβ42 and *s*-Aβ42 using ANA-SA5 (Figure 9). These data indicated that ANA-SA5 can cleave *s*-Aβ42, which is thought to be similar to the senile plaques. This implied that ANA-SA5, ANA-TA9, and JAL-TA9 could potentially be useful for the treatment of AD.

### 3.4. Effect of ANA-SA5 on the Growth of A549 Cells

Cytotoxicity is one of the most important events in clinical settings. Therefore, the cytotoxicity of ANA-SA5 was evaluated using A549 cells, which are one of the cells that can be used for drug metabolism research. ANA-SA5 (0.2 mM) did not show a significant inhibitory effect on the growth of A549 cells compared to the FLAG peptide, which was used as a peptide control. In contrast, the chemotherapeutic agent CDDP, which was used as the positive control, inhibited the growth of A549 cells (Figure 10).

## 4. Discussion

In this study, we found that a 5-mer peptide, ANA-SA5, cleaves both *a*-Aβ42 and *s*-Aβ42 by serine-protease-like activity. In both cases, ANA-SA5 cleaved at numerous positions on *s*-Aβ42 and *a*-Aβ42 that were different from each other (Figure 9). Previous reports examining the cytotoxicity of fragment peptides of various lengths derived from Aβ42 have reported that Aβ25—35, Aβ1—36, 1—39, and 1—42 are cytotoxic fragment peptides. Other fragments from the N-terminus and middle region have been shown not to exhibit cytotoxicity [24]. In this study, it was revealed that ANA-SA5 cleaved Aβ42, and the cleaved fragment peptides do not contain fragment peptides that have been reported to be cytotoxic. This suggests that the Aβ42 fragment peptide cleaved by ANA-TA9 does not exhibit cytotoxicity.

The specificity of this cleavage site has not been determined. It is generally believed that enzymes exhibit cleavage site specificity. Chymotrypsin, a representative proteolytic enzyme, hydrolyzes peptides at the carboxyl groups of aromatic amino acids. Chymotrypsin contains a hydrophobic pocket as a substrate-binding site. An aromatic amino acid, which is highly hydrophobic, binds to the hydrophobic pocket, and its carboxy group approaches the active center of chymotrypsin, causing its degradation [23]. Cleavage site specificity is believed to occur because of this mechanism. This was also true for other enzymes, and the specificity of the cleavage point was determined by the substrate-binding site. However, ANA-SA5, composed of five amino acid residues, is a very small molecule that lacks a substrate-binding site, similar to the enzymes mentioned above. This is thought to be why it did not exhibit any cut-point characteristics. This indicates that the Catalytides are very different from the lock-and-key relationships expressed by the common enzymes.

We assumed that the difference in the cleavage sites of ANA-SA5 for *s*-Aβ42 and *a*-Aβ42 is due to the interaction with the substrate and the substrate stereochemistry and can change depending on the stereo-structure of Aβ42. However, the mechanism underlying the hydrolytic activity remains unclear. We further hypothesized that the compact structure of ANA-SA5 allows it to invade the inner space of aggregated/oligomerized substrates and cleave them.

The hydrolytic activity mechanism of ANA-SA5 was elucidated through structural analysis, utilizing the MM2 and MMFF94 parameters. These findings led us to propose a cleavage mechanism for ANA-SA5 (Figure 11). Initially, the nitrogen atom at the N-terminus extracts a proton from the hydroxyl group of serine, forming an oxyanion. This oxyanion, acting as a nucleophile, proceeds to attack the peptide bonds of the substrate. Following this, the N-terminus (-NH_2_) donates a proton to the nitrogen of the amide substrate. Lastly, water attacks the ester bond between the substrate and Ser (Figure 11). This proposed mechanism is consistent with the well-established chemical process of serine protease catalysis [23].

For clinical use, two important events must be clarified. The first are side effects, including the toxicity or cleavage of enzymes against various major proteins. In this study, ANA-SA5 did not show any effect on the growth of A549 cells (Figure 10), similar to that of JAL-TA9 and ANA-TA9 cells. In addition, we reported that two types of Catalytides, JAL-TA9 and ANA-TA9, did not show proteolytic activity against five proteins: γ-globulin, rabbit immunoglobulin G, cytochrome C, and lysozyme, as well as HSA [6]. ANA-SA5 is a component of ANA-TA9, and the proteolytic properties of ANA-SA5 are like those of ANA-TA9 and JAL-TA9, suggesting that ANA-SA5 does not cleave these native proteins and is non-toxic [3,5,6,21].

Second, basic information regarding the pharmacokinetics of ANA-SA5, such as its stability, degradation, absorption, and brain delivery, is important. Recently, it has become widely accepted that peptides and proteins can be transported to the central nervous system directly from the nose [25,26,27,28,29,30,31,32] because of several advantages over oral or intraperitoneal administration, including the non-invasiveness, self-administration, shorter onset time of the effect, and higher bioavailability, owing to the avoidance of hepatic first-pass metabolism. Moreover, intranasal drug administration has gained increasing interest as an application site for drugs, particularly peptides, for systemic delivery. Recently, we revealed that ANA-TA9 could be delivered to the brain via nasal administration [7]. Based on these data, we expect that ANA-SA5 can be efficiently delivered to the brain via nasal administration, similar to ANA-TA9, compared to other routes.

## 5. Conclusions

This study demonstrated that ANA-SA5, the analogous 5-mer peptide of ANA-TA9 derived from the Box A region of the ANA/BTG3 protein, cleaves both *a*-Aβ42 and *s*-Aβ42 with serine-protease-like activity, which is similar to the proteolytic activity of ANA-TA9 and JAL-TA9. We should examine whether Aβ42 fragment peptides which are generated by ANA-SA5 from Aβ42 showed cytotoxicity or not in the future. Even with this limitation, we concluded that ANA-SA5 is an attractive candidate for clinical use in AD treatment and prevention, as well as ANA-TA9 and JAL-TA9, previously identified as Catalytides.

## 6. Patents

T. Yamamoto and T. Akizawa, 2016, NOVEL HYDROLASE-LIKE PEPTIDE AND ITSUSE, US62/275,599.

## Figures and Tables

**Figure 1 biomolecules-14-00586-f001:**
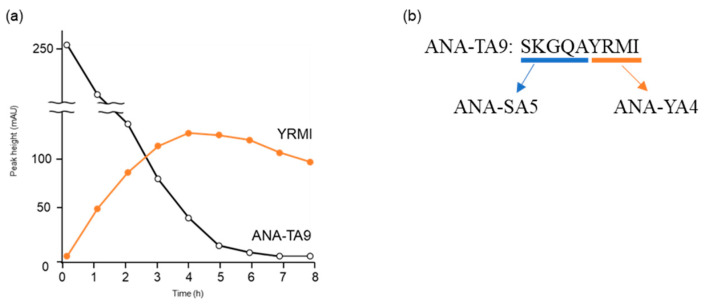
Time-dependence analysis of autoproteolytic activity of ANA-TA9. (**a**) The reaction mixture was analyzed on an analytical HPLC, and peak height was plotted at an absorbance of 220 nm. (**b**) Sequences of ANA-TA9, ANA-SA5, and ANA-YA4.

**Figure 2 biomolecules-14-00586-f002:**
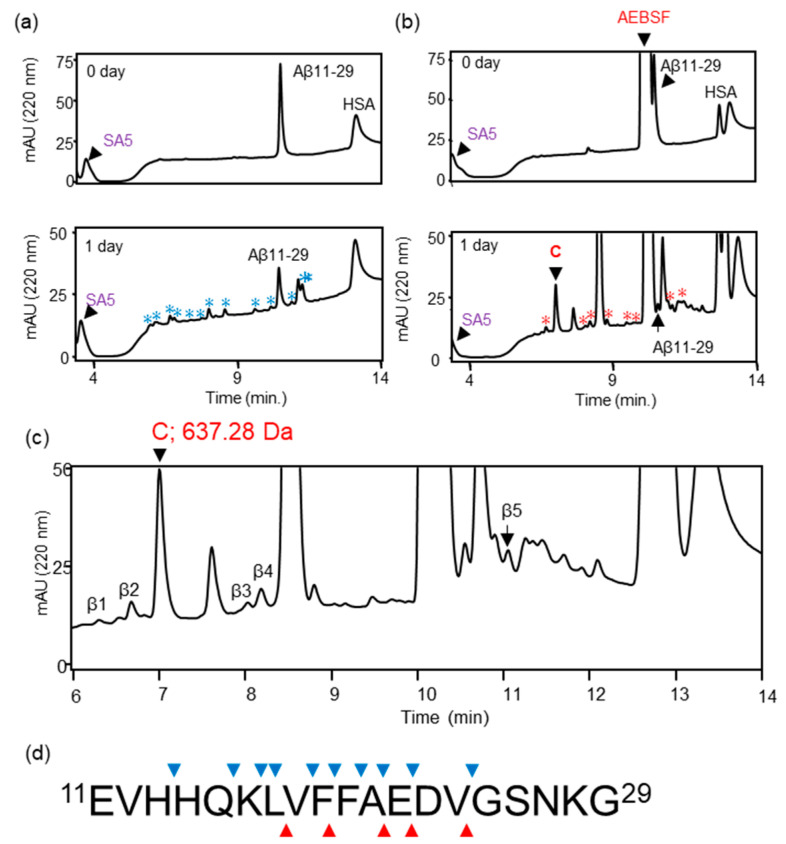
Inhibitory effect of AEBSF on the proteolytic activity of ANA-SA5. Aβ11—29 (50 μM) was incubated with ANA-SA5 (200 μM) in PBS at 37 °C with or without AEBSF. (**a**) Chromatogram of Aβ11—29 co-incubated with ANA-SA5. The peaks marked with a blue star (*) were peaks corresponding to fragment peptide of Aβ11—29. (**b**) Chromatogram of Aβ11—29 co-incubated with ANA-SA5 in the presence of AEBSF. The peaks marked with a red star (*) are not peptides. For both (**a**) and (**b**), ten microliters of the reaction mixture were injected into the analytical HPLC system. The upper figure represents the results at day 0, and the lower figure represents the results at day 1. (**c**) Five peak fractions were identified as peptide fragments derived from Aβ11—29. Twenty microliters of the reaction mixture were injected into the analytical HPLC system. (**d**) Cleavage sites on Aβ11—29 by ANA-SA5 (▼ in the absence of AEBSF; ▲ in the presence of AEBSF).

**Figure 3 biomolecules-14-00586-f003:**
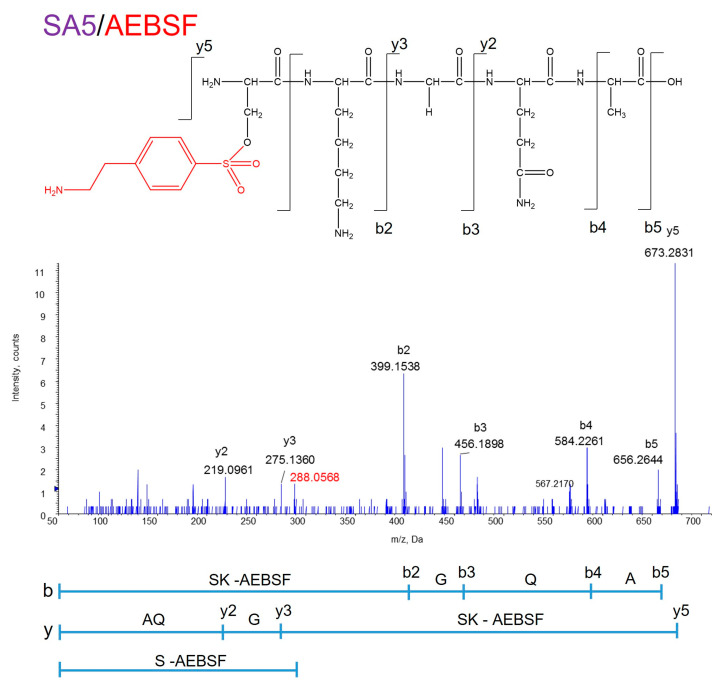
MS/MS analysis of the ANA-SA5 and AEBSF complex. The complex of AEBSF and the serine residue of ANA-SA5 was identified with a molecular weight of 288.0568. The production obtained by fragmenting the precursor ion 673.28 by collision-induced dissociation was read. The reading method from the C-terminus is described as b, and the reading method from the N-terminus is described as y.

**Figure 4 biomolecules-14-00586-f004:**
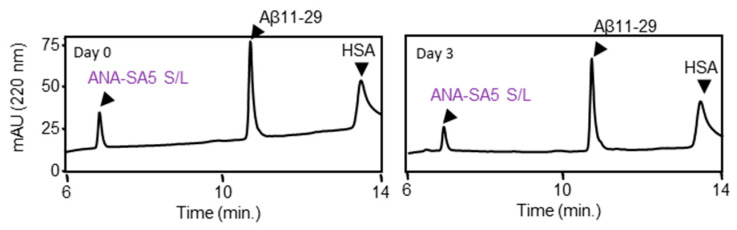
Evaluation of the proteolytic activity of ANA-SA5 S/L. Aβ11—29 (50 μM) was incubated with ANA-SA5 S/L (200 μM) in PBS at 37 °C. Ten μL of the reaction mixture was injected into the analytical HPLC system.

**Figure 5 biomolecules-14-00586-f005:**
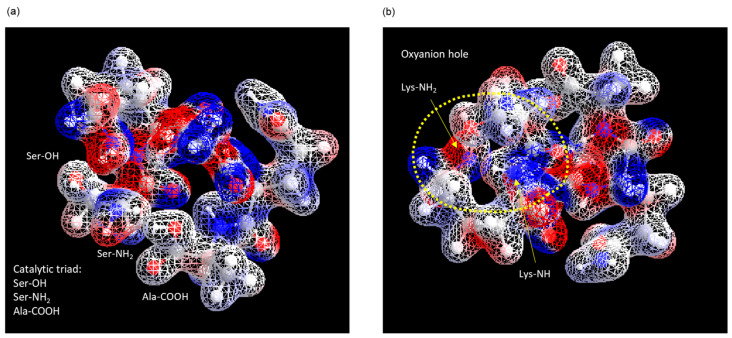
Computer modeling using MM2 and MMFF94 parameters of ANA-SA5. Simulated structure centered on (**a**) catalytic triad and (**b**) oxyanion hole.

**Figure 6 biomolecules-14-00586-f006:**
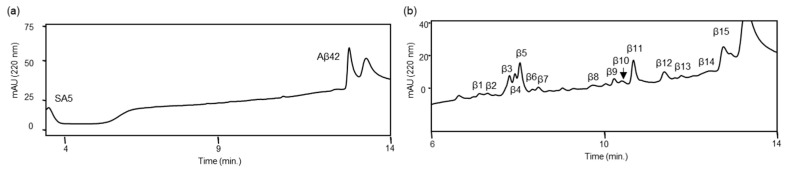
Cleavage reaction of the authentic soluble form Aβ42 (*a*-Aβ42) by ANA-SA5. Aβ42, with a final concentration of 0.05 mM, was incubated with HSA (final concentration: 0.025% *w*/*v*) in phosphate-buffered saline at 37 °C in the presence of ANA-SA5 (final concentration 0.2 mM). Initially, 10 μL of the reaction mixture was analyzed using analytical HPLC (**a**). After 7 days, 20 μL of the reaction mixture was injected, and all new peaks were collected (**b**).

**Figure 7 biomolecules-14-00586-f007:**
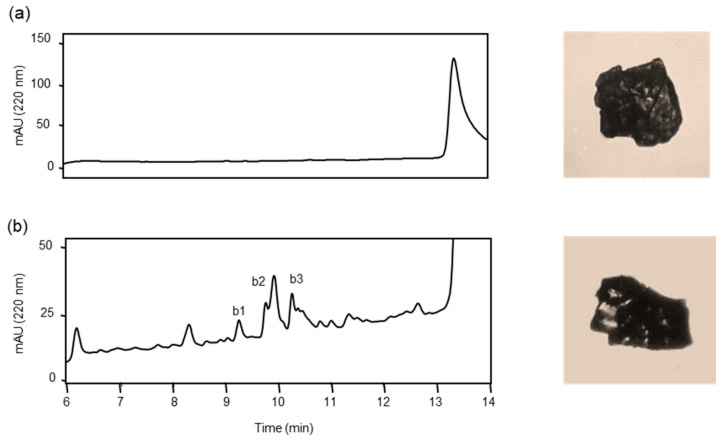
Cleavage reaction of solid form Aβ42 (*s*-Aβ42). *s*-Aβ42 (0.3 mg) was incubated with HSA (final concentration: 0.025% *w*/*v*) in PBS at 37 °C. (**a**) Initially, 10 μL of the reaction mixture was analyzed using an analytical HPLC system and a photograph of *s*-Aβ42. (**b**) After 7 days, 40 μL of the reaction mixture was injected, and all new peaks were collected and analyzed using the analytical HPLC system, and a photograph of *s*-Aβ42 was obtained.

**Figure 8 biomolecules-14-00586-f008:**
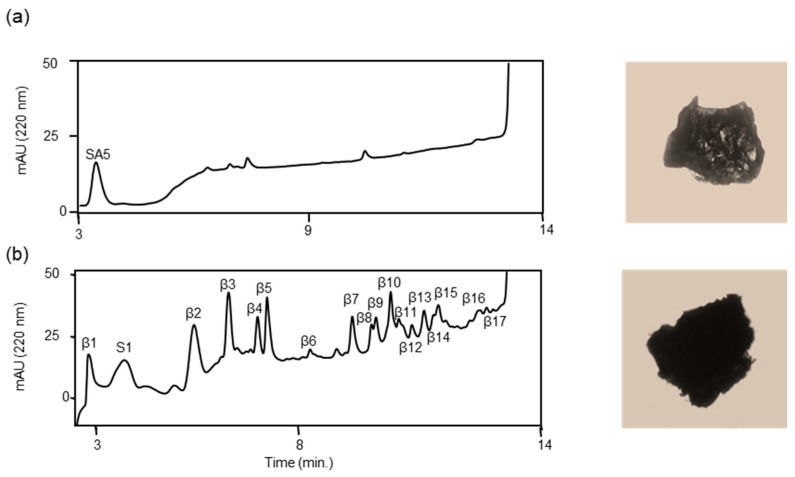
Cleavage reaction of the solid form Aβ42 (*s*-Aβ42) by ANA-SA5. *s*-Aβ42 (0.3 mg) was incubated with HSA (final concentration 0.025% *w*/*v*) in phosphate-buffered saline at 37 °C with ANA-SA5 (final concentration 1.0 mM). (**a**) Initially, 10 μL of the reaction mixture was analyzed using the analytical HPLC system and a photograph of *s*-Aβ42. (**b**) After 7 days, 40 μL of the reaction mixture was injected, and all the new peaks were collected and analyzed by MS, and a photograph of *s*-Aβ42 was obtained.

**Figure 9 biomolecules-14-00586-f009:**
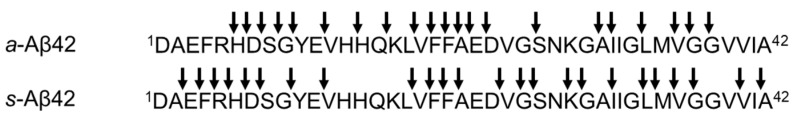
Summary of the cleavage sites on *a*-Aβ42 and *s*-Aβ42 by ANA-SA5. Black arrow indicates the location of the cleavage sites by ANA-TA9.

**Figure 10 biomolecules-14-00586-f010:**
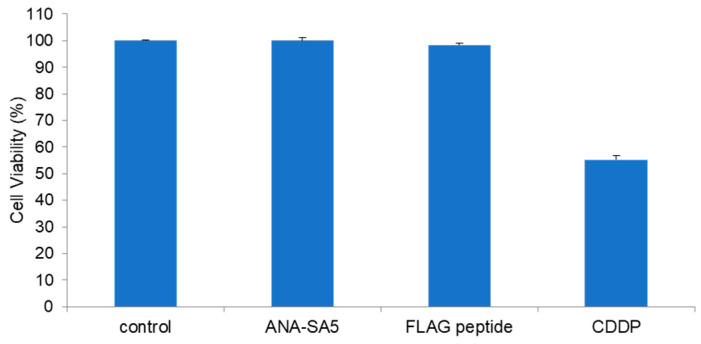
The relative viability of A549 cells cultured in the presence or absence of 0.2 mM peptides (ANA-SA5, FLAG peptide) for 72 h. CDDP (4 μM), an anticancer drug, was used as the positive control. Cell viability was determined using a WST-8 assay. Each bar represents the mean ± SE (*n* = 4).

**Figure 11 biomolecules-14-00586-f011:**
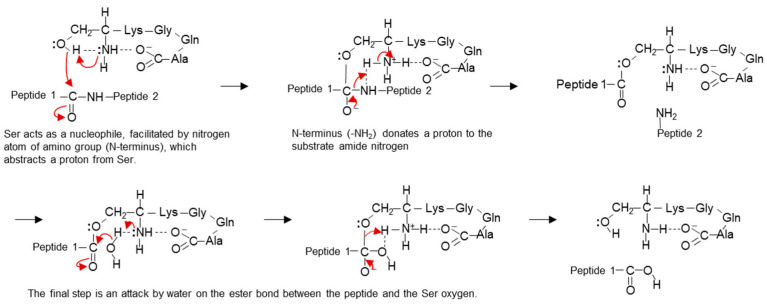
Proposed cleavage mechanism of ANA-SA5.

**Table 1 biomolecules-14-00586-t001:** Fragment peptides derived from Aβ11—29 cleaved by ANA-SA5 in the presence of AEBSF. Cleavage sites were determined using a Qstar Hybrid LC-MS/MS system (ABI).

Peak	Fragment	Calculated Mass	Observed Mass
β1	DVGSNKG	676.3260	676.3221
	GSNKG	461.2307	461.2234
β2	EDVGSNKG	805.3686	805.3601
β3	EVHHQKL	890.4842	890.5015
β4	FAEDVGSNKG	1023.4741	1023.4890
β5	EVHHQKLVFFAE	1482.76	1482.7860

**Table 2 biomolecules-14-00586-t002:** The distances of amino acids forming catalytic triad by structural analysis.

	Length (Å)
Ser (-OH)-Ser(-NH_2_)	2.26
Ala (-COOH)-Ser(NH_2_)	2.03

**Table 3 biomolecules-14-00586-t003:** Mass spectrometry (MS) analysis of the reaction mixture of *a*-Aβ42 and ANA-SA5.

Peak	Fragment	Calculated Mass	Observed Mass
β1	AEDVGSNKG	876.4057	876.4001
	AEDVG	490.2143	490.2314
β2	AEDVGSNKGA	947.4428	947.4432
β3	HQKL	525.3143	525.3073
	YEVH	547.2511	547.2429
	F	166.0862	166.0736
β4	DAEFRH	774.3529	774.3491
β5	DSGYEVH	806.3315	806.3282
	SGYEVH	691.3046	691.3023
	GYEVH	604.2725	604.2654
β6	DAEFR	637.2940	637.2875
β7	HQKLV	624.3827	624.3879
	FAEDVGSNKGA	1094.5112	1094.5014
β8	GGVVIA	515.3187	515.3027
	GVVIA	458.2973	458.2883
	FFAEDVGSNKGA	1241.5796	1241.5872
β9	FFAE	513.2343	513.2232
	VHHQ	520.2626	520.3233
β10	HQKLVF	771.4512	771.4414
β11	AIIG	373.2445	373.2334
β12	VFFA	483.2602	483.2494
	AEDVGSNKGAIIGLM	1474.7570	1474.7415
β13	AIIGLM	617.3691	617.3527
	HQKLVFF	918.5196	918.5002
β14	IIGLM	546.3320	546.3109
β15	Aβ42	4512.2768	4512.2800

**Table 4 biomolecules-14-00586-t004:** MS analyses of the reaction mixture of *s*-Aβ42.

Peak	Fragment	Calculated Mass	Observed Mass
b1	GGVVIA	515.3187	515.4486
	VVIA	401.2758	401.3619
b2	GVVIA	458.2973	458.4021
b3	VGGVVIA	614.3871	614.5443

**Table 5 biomolecules-14-00586-t005:** MS analyses of the reaction mixture of *s*-Aβ42 and ANA-SA5.

Peak	Fragment	Calculated Mass	Observed Mass
S1	SKGQA	490.2620	490.2721
β1	QK	275.1714	275.1445
β2	SGYE	455.1772	455.0973
β3	R	175.1189	175.1210
β4	IIGLM	546.3320	546.3343
β5	GAII or AIIG	373.2445	373.2331
	AII	316.2231	316.2004
β6	EFRH	588.2888	588.3154
β7	KGAIIG	558.3609	558.3452
β8	GGVVIA	515.3187	515.3301
β9	VGSNKGAIIG	915.5258	914.6014
β10	YEVHHQKL	1053.5476	1053.5873
	FAEDVGSNK	966.4527	966.5642
β11	VFF	412.2231	412.2643
β12	KGAIIGL	671.4450	671.4257
β13	FRH	459.2463	459.2905
β14	KGAIIGLM	802.4855	802.4546
β15	FFAEDVG	784.3512	784.5042
β16	LMVGGVVI	787.4746	787.4581
β17	EVHHQKLVFFAED	1598.7961	1598.9801
	NKGAIIGLMVGGV	1228.7082	1228.7654

## Data Availability

The data and materials in this article are available from the corresponding authors upon reasonable request.

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
