# Peer review of "SKGQA, a Peptide Derived from the ANA/BTG3 Protein, Cleaves Amyloid-β with Proteolytic Activity"

_biomolecules, 2024, doi:10.3390/biom14050586_

Round 1

Reviewer 1 Report

Comments and Suggestions for Authors

Dear Sirs,

The manuscript “SKGQA, a peptide derived from the ANA/BTG3 protein, cleaves amyloid-β with proteolytic activity” describes the 5-mer ANA-SA5 proteolytic properties towards Aβ peptides, highlighting its possible use as therapeutics in AD treatment and prevention.

The work is interesting and well conducted, particularly concerning the characterization of the proteolytic activity of ANA-SA5, providing details on the identification of the active center, and proteolysis of two forms of Aβ42. However, the toxicity of the Aβ42 fragments was not assessed for their toxicity, namely in cell culture.

Nevertheless, doubts and questions arose while reading the manuscript. The following comments are believed to be pertinent and may be important to improve the work:

Material and Methods section

1-     For the chemical synthesis of peptides, there is no reference to any procedure for the “authentic” soluble form, and it is not clear if both forms are obtained through any common process.

2-     Please provide some information on the cell line used (A549 cells), the rationale behind its selection, and how these cells fit the objectives of the work.

Section 3.1 of Results:

3-     In the following sentence (page 4, lines 165-167): “The reaction mixture was loaded onto an analytical HPLC system. In the presence of AEBSF, both ANA-SA5 and Aβ11-29 were initially identified as a single peak (Figure 1a).”

Do you mean a single peak for each? (One for ANA-SA5 and another for Aβ11-29, as Figure 1a suggests.

4-     Regarding results displayed in Figure 1a, the respective experiment should have been conducted with and without AEBSF, regardless of previous experiments showing that ANA-SA5 cleaves Aβ11-29 in 13 peptide fragments. It is important to reproduce results using the current reagents and conditions and to demonstrate that indeed AEBSF partially inhibits ANA-SA5 activity, by comparing the two situations.

5-     Figure 1a is composed of 2 graphs; the upper graph is at day 0, and the lower graph is at day 1. Both of them should be in the presence of AEBSF, right? But the lower graph does not show that information, as the upper graph does.

The legend in Figure 1 also misses information regarding (a), since the legend does not discriminate at all the existence of the 2 graphs at different time points.

6-     On page 4, lines 176-178 are difficult to understand:

“…peaks marked by C were identified as 637.28 Da.” First, “peaks” is plural, and only one value of mass is given; second, does the value 637.28 Da appear anywhere in Figure 1a or b? Clarify in the text.

And next: “The molecular weight was the same as that obtained when ANA-SA5 and AEBSF were used in combination.”

Isn’t the entire experiment in the presence of AEBSF? What is the comparison that the authors are doing here?

Finally, “Thus, we believe that AEBSF and ANA-SA5 are combined”. Do authors mean binding of ANA-SA5 to AEBSF?

7-     In the last paragraph of page 4, the reference to Figure 2 should appear earlier to make clear to which results the text refers. And in line 186, the reference to “C5” is not understandable. What is C5?

8-     On page 10, line 291: the authors state that solid Aβ is thought to be similar to toxic Aβ42 oligomers. Oligomers are toxic, soluble Aβ species, and solid Aβ even considering the description made by authors, is insoluble. In this sense, solid Aβ seems to be closer to Aβ present in senile plaques.

9-     A significant limitation of this work, especially since viability tests were carried out, is that the toxicity of the fragments of Aβ42 proteolysis by ANA-SA5 was not evaluated.

As the authors state, cytotoxicity is one of the most important events in clinical settings. To conclude on the use of ANA-SA5 as therapy in AD, the generated Aβ42 fragments should be assessed for their cytotoxicity, since these fragments may still be toxic.

Minor comments:

-        Page 3, line 124: “A portion of the reaction mixture (20 μL for α-Aβ42 and Aβ11-29; 40 μL for s-Aβ42)”, should be a-Aβ42.

-        Page 10, line 289: shouldn’t it be Table 5 instead of Table 4?

-        About the effect of ANA-SA5 on the growth of A549 cells, the authors wrote (page 12, lines 307-309) “Therefore, we examined the effect of ANA-SA5 on the growth of A549 cells before the in vivo experiments (Figure 9).” This is somehow misleading, as no in vivo studies are presented, and could be changed to:

Therefore, we examined the effect of ANA-SA5 on the growth of A549 cells (Figure 9).”

Author Response

Thank you very much for providing important comments. We are thankful for the time and energy you expended. Our responses to the comments are as follows:

Material and Methods section

  1. For the chemical synthesis of peptides, there is no reference to any procedure for the “authentic” soluble form, and it is not clear if both forms are obtained through any common process

Response: Thank you for pointing out. We noticed there are no citations for how authentic soluble forms were prepared. We have added the sentence (page 3, lines 125-126): “Authentic soluble form Aβ42 was purchased from peptide institution”. 

  1. Please provide some information on the cell line used (A549 cells), the rationale behind its selection, and how these cells fit the objectives of the work.

Response: Thank you for pointing out. We have added where we obtained from (page 4, lines 147-148) and why we chose the A549 cells (page 13, lines 318-320).

Section 3.1 of Results:

  1. In the following sentence (page 4, lines 165-167): “The reaction mixture was loaded onto an analytical HPLC system. In the presence of AEBSF, both ANA-SA5 and Aβ11-29 were initially identified as a single peak (Figure 1a).”

Do you mean a single peak for each? (One for ANA-SA5 and another for Aβ11-29, as Figure 1a suggests.

Response: It is exactly as you said, “single peak for each”. We have rewritten the sentence (page 4, line 177) and inserted what peaks are corresponding (page 5, Figure 2a) to be more in line with your comments.

  1. Regarding results displayed in Figure 1a, the respective experiment should have been conducted with and without AEBSF, regardless of previous experiments showing that ANA-SA5 cleaves Aβ11-29 in 13 peptide fragments. It is important to reproduce results using the current reagents and conditions and to demonstrate that indeed AEBSF partially inhibits ANA-SA5 activity, by comparing the two situations.

Response: We have inserted the cleavage sites on Aβ11-29 by ANA-SA5 without AEBSF condition (Figure 2c, page 5) to make it clear to compare them with and without AEBSF.

  1. Figure 1a is composed of 2 graphs; the upper graph is at day 0, and the lower graph is at day 1. Both of them should be in the presence of AEBSF, right? But the lower graph does not show that information, as the upper graph does.

The legend in Figure 1 also misses information regarding (a), since the legend does not discriminate at all the existence of the 2 graphs at different time points.

Response: Thank you for pointing out. We have added information about upper and lower meaning (page 5, line 206).

  1. On page 4, lines 176-178 are difficult to understand:

“…peaks marked by C were identified as 637.28 Da.” First, “peaks” is plural, and only one value of mass is given; second, does the value 637.28 Da appear anywhere in Figure 1a or b? Clarify in the text.

Response: Thank you for pointing out. First, “peaks were rewritten to peak.” Second, “the peak marked by C” appeared in the lower graph of Figure 2a, and we also added C and 637.28 Da in Figure 2b.

And next: “The molecular weight was the same as that obtained when ANA-SA5 and AEBSF were used in combination.”

Isn’t the entire experiment in the presence of AEBSF? What is the comparison that the authors are doing here?

Finally, “Thus, we believe that AEBSF and ANA-SA5 are combined”. Do authors mean binding of ANA-SA5 to AEBSF?

Response: We meant that the 637.28 Da were predicted to be the molecular weight which are combination of AEBSF and ANA-SA5. We have rewritten the sentence (page 5, lines 187-188).

  1. In the last paragraph of page 4, the reference to Figure 2 should appear earlier to make clear to which results the text refers. And in line 186, the reference to “C5” is not understandable. What is C5?

Response: We inserted the reference to Figure earlier as you mentioned (page 5, line 192). C5 is miswritten, so we corrected it to y5.

  1. On page 10, line 291: the authors state that solid Aβ is thought to be similar to toxic Aβ42 oligomers. Oligomers are toxic, soluble Aβ species, and solid Aβ even considering the description made by authors, is insoluble. In this sense, solid Aβ seems to be closer to Aβ present in senile plaques.

Response: Thank you for providing the insight. We have reflected this comment on page 11, line 302.

  1. A significant limitation of this work, especially since viability tests were carried out, is that the toxicity of the fragments of Aβ42 proteolysis by ANA-SA5 was not evaluated.

As the authors state, cytotoxicity is one of the most important events in clinical settings. To conclude on the use of ANA-SA5 as therapy in AD, the generated Aβ42 fragments should be assessed for their cytotoxicity, since these fragments may still be toxic.

Response: We understood the limitation of this study as you pointed out. So, we inserted the limitation in the Conclusions section (page 15, lines 387-388).

Minor comments:

-Page 3, line 124: “A portion of the reaction mixture (20 μL for α-Aβ42 and Aβ11-29; 40 μL for s-Aβ42)”, should be a-Aβ42.

-Page 10, line 289: shouldn’t it be Table 5 instead of Table 4?

-About the effect of ANA-SA5 on the growth of A549 cells, the authors wrote (page 12, lines 307-309) “Therefore, we examined the effect of ANA-SA5 on the growth of A549 cells before the in vivo experiments (Figure 9).” This is somehow misleading, as no in vivo studies are presented, and could be changed to:

“Therefore, we examined the effect of ANA-SA5 on the growth of A549 cells (Figure 9).”

Response: We have corrected all minor comments (page 3, line 133; page 11, line 300; page 13, line 318-320).

Reviewer 2 Report

Comments and Suggestions for Authors

The research article entitled "SKGQA, a peptide derived from the ANA/BTG3 protein cleaves amyloid-β with proteolytic activity" is focused on the development of the peptide for the proteolysis of Amyloid Bata in Alzheimer’s disease (AD). The authors reported the nine amino acids containing synthetic peptides. The peptide synthesized was characterized and evaluated for its proteolytic activity. The manuscript is well-written and may be accepted after minor revision. 

Minor Comments-

1. Why authors used only HPLC for the characterization and identification of proteolytic activity? MASS should also be used for better understanding.

2. The MASS/NMR spectra of the synthesized peptide should also be included in the supplementary and the same should be discussed in the manuscript.   

3. Designing approach for the identification of the developed peptide should be illustrated and discussed in the manuscript for better understanding by the audience. 

Author Response

Thank you very much for providing important comments. We are thankful for the time and energy you expended. Our responses to the comments are as follows:

  1. Why authors used only HPLC for the characterization and identification of proteolytic activity? MASS should also be used for better understanding.

Response: We use HPLC to analyze the peaks, and then we use Mass to analyze them. The results of the analysis are shown in Table 1 and 3-5, and we also wrote how we determined the cleavage sites (page 3, lines 138-140)

  1. The MASS/NMR spectra of the synthesized peptide should also be included in the supplementary, and the same should be discussed in the manuscript.

Response: Thank you for pointing this out. We inserted the supplementary on page 3, lines 122-126.

  1. Designing approach for the identification of the developed peptide should be illustrated and discussed in the manuscript for better understanding by the audience.

Response: Thank you for providing the insights. We inserted the Figure in the introduction section and wrote how we apported on Page 2, lines 72-75.

At first, I thought about writing the ideas in the discussion section. However, we intentionally chose to include a designing approach in the introduction for better understanding by the audience for the following reasons:

The reason why ANA-SA5 and ANA-YA4 were synthesized from ANA-TA9 (how we found ANA-TA9 composed of two Catalytides) and we also evaluate the enzyme activity of ANA-TA9, ANA-SA5, and ANA-YA4 against Aβ42 fragment peptide was described in our previous paper. Therefore, in this paper, we aim to discuss and highlight that Catalytide ANA-SA5, which was synthesized from ANA-TA9, cleaves the Aβ42 peptide and has the potential to be a therapeutic agent for Alzheimer’s disease.

Round 2

Reviewer 1 Report

Comments and Suggestions for Authors

Dear Sirs,

The authors have clarified several points in the manuscript, but important concerns still exist as the authors did not answer properly:

1-     For the chemical synthesis of peptides, there is no reference to any procedure for the “authentic” soluble form, and it is not clear if both forms are obtained through any common process.

The authors added the source of the peptide (in the current version of the manuscript, they added “…institute”, and not institution, actually) but not how it was handled.

Was the peptide dissolved in HFIP or other? The stock was in DMSO or other? Concentration of the stock solution? Was it prepared fresh?

2-     Figure 1a is composed of 2 graphs; the upper graph is at day 0, and the lower graph is at day 1. Both of them should be in the presence of AEBSF, right? But the lower graph does not show that information, as the upper graph does.

Figure 2A is still confusing. The Term “+ AEBSF” (in red) is inside the rectangle of the upper chromatogram, but not in the lower.

In my opinion, and because the legend is clear concerning the presence of AEBSF, the “+AEBSF” should be removed. Or find a way to clearly indicate that both chromatograms are in the presence of AEBSF.

Finally, 2 of the main concerns raised were not really addressed by the authors:

1-     Regarding results displayed in Figure 1a, the respective experiment should have been conducted with and without AEBSF, regardless of previous experiments showing that ANA-SA5 cleaves Aβ11-29 in 13 peptide fragments. It is important to reproduce results using the current reagents and conditions and to demonstrate that indeed AEBSF partially inhibits ANA-SA5 activity, by comparing the two situations.

2-     A significant limitation of this work, especially since viability tests were carried out, is that the toxicity of the fragments of Aβ42 proteolysis by ANA-SA5 was not evaluated.

Round 3

Reviewer 1 Report

Comments and Suggestions for Authors

Dear sirs,

It is my opinion that the authors have now addressed all concerns and significantly improved the work, and thus I endorse the publication of this manuscript.